# β-Glucocerebrosidase Deficiency Activates an Aberrant Lysosome-Plasma Membrane Axis Responsible for the Onset of Neurodegeneration

**DOI:** 10.3390/cells11152343

**Published:** 2022-07-29

**Authors:** Giulia Lunghi, Emma Veronica Carsana, Nicoletta Loberto, Laura Cioccarelli, Simona Prioni, Laura Mauri, Rosaria Bassi, Stefano Duga, Letizia Straniero, Rosanna Asselta, Giulia Soldà, Alessio Di Fonzo, Emanuele Frattini, Manuela Magni, Nara Liessi, Andrea Armirotti, Elena Ferrari, Maura Samarani, Massimo Aureli

**Affiliations:** 1Department of Medical Biotechnology and Translational Medicine, University of Milan, 20054 Milan, Italy; giulia.lunghi@unimi.it (G.L.); emma.carsana@unimi.it (E.V.C.); nicoletta.loberto@unimi.it (N.L.); laura.cioccarelli@hotmail.it (L.C.); simona.prioni@unimi.it (S.P.); laura.mauri@unimi.it (L.M.); rosaria.bassi@unimi.it (R.B.); 2Department of Biomedical Sciences, Humanitas University, 20090 Milan, Italy; stefano.duga@hunimed.eu (S.D.); letizia.straniero@humanitasresearch.it (L.S.); rosanna.asselta@hunimed.eu (R.A.); giulia.solda@hunimed.eu (G.S.); 3Humanitas Clinical and Research Center—IRCCS, Via Manzoni 56, 20072 Milan, Italy; 4IRCCS Foundation Ca’ Granda Ospedale Maggiore Policlinico, Dino Ferrari Center, Neuroscience Section, Department of Pathophysiology and Transplantation, University of Milan, 20122 Milan, Italy; alessio.difonzo@policlinico.mi.it (A.D.F.); emanuele.frattini@unimi.it (E.F.); manuelamagni90@gmail.com (M.M.); 5Analytical Chemistry Facility, Fondazione Istituto Italiano di Tecnologia, Via Morego 30, 16163 Genoa, Italy; nara.liessi@iit.it (N.L.); andrea.armirotti@iit.it (A.A.); 6Department of Pharmacological and Biomolecular Sciences, University of Milan, 20133 Milan, Italy; elena.ferrari2@unimi.it; 7Department of Cell Biology and Infection, Institut Pasteur, 75015 Paris, France; maura.samarani@pasteur.fr

**Keywords:** GBA1, glucosylceramide, Gaucher disease, lysosomes, plasma membrane, lipid rafts

## Abstract

β-glucocerebrosidase is a lysosomal hydrolase involved in the catabolism of the sphingolipid glucosylceramide. Biallelic loss of function mutations in this enzyme are responsible for the onset of Gaucher disease, while monoallelic β-glucocerebrosidase mutations represent the first genetic risk factor for Parkinson’s disease. Despite this evidence, the molecular mechanism linking the impairment in β-glucocerebrosidase activity with the onset of neurodegeneration in still unknown. In this frame, we developed two in vitro neuronal models of β-glucocerebrosidase deficiency, represented by mouse cerebellar granule neurons and human-induced pluripotent stem cells-derived dopaminergic neurons treated with the specific β-glucocerebrosidase inhibitor conduritol B epoxide. Neurons deficient for β-glucocerebrosidase activity showed a lysosomal accumulation of glucosylceramide and the onset of neuronal damage. Moreover, we found that neurons react to the lysosomal impairment by the induction of their biogenesis and exocytosis. This latter event was responsible for glucosylceramide accumulation also at the plasma membrane level, with an alteration in lipid and protein composition of specific signaling microdomains. Collectively, our data suggest that β-glucocerebrosidase loss of function impairs the lysosomal compartment, establishing a lysosome–plasma membrane axis responsible for modifications in the plasma membrane architecture and possible alterations of intracellular signaling pathways, leading to neuronal damage.

## 1. Introduction

β-glucocerebrosidase (GCase) is a lysosomal hydrolase encoded by *GBA1* gene that catalyzes the hydrolysis of the glycosphingolipid (GSL) glucosylceramide (GlcCer) to ceramide and glucose. The severe deficiency of GCase results in the accumulation of GlcCer, which is classically linked to the multi-organ clinical manifestations typical of Gaucher disease (GD), the most common lysosomal storage disorder (LSD) [1,2].

Mutations in *GBA1* are associated with an increased incidence of Parkinson’s disease (PD), both in GD patients and in heterozygous carriers, representing the main genetic risk factor for the development of PD [3,4,5,6,7]. Interestingly, a deficiency of GCase activity in the substantia nigra pars compacta has been demonstrated not only in PD patients carrying *GBA1* mutations, but also in PD subjects with a wild-type *GBA1* [8,9]. Furthermore, Murphy and co-workers analyzed brain tissues from sporadic PD patients without *GBA1* mutations and discovered that deficits in GCase are positively correlated with the levels of accumulated alpha-synuclein. Interestingly, the accumulation of alpha-synuclein within lysosomes has been associated with a chronic reduction in lysosomal function [10,11].

Different theories have been formulated to explain how and to which extent GCase deficiency contributes to neurodegeneration [12]. It has been hypothesized that a central role could be played by the endolysosomal compartment where GlcCer accumulation could directly influence the abnormal lysosomal storage of alpha-synuclein oligomers, thus resulting in a further inhibition of GCase activity. These findings suggest the establishment of a positive feedback loop between GCase deficiency and alpha-synuclein accumulation, which may lead to a self-propagation of the disease [13]. In line with this evidence, the rescue of GCase activity ameliorates the neurodegenerative phenotype, suggesting a possible direct involvement of the accumulated GlcCer in neurodegeneration [14,15]. Despite these findings, the molecular mechanism linking GCase impairment with neurodegeneration remains unclear and still debated in the scientific community. A major obstacle in the understanding of the molecular mechanism linking GCase impairment with neurodegeneration is represented by the lack of suitable in vitro and in vivo models.

In this study, we developed two different neuronal in vitro models of GCase deficiency represented by murine cerebellar granule neurons (CGN) and human-induced pluripotent stem cell (iPSC)-derived dopaminergic (DA) neurons, both treated with conduritol B epoxide (CBE) a selective inhibitor of GCase activity [16]. Our data highlight the existence of a pathogenic lysosome–plasma membrane (PM) axis that participates in the onset of neurodegeneration upon GCase deficiency, thus adding a new piece to the already multifaceted research on GCase-related pathologies.

## 2. Materials and Methods

### 2.1. Materials

Commercial chemicals were of the highest purity available, common solvents were distilled before use and water was doubly distilled in a glass apparatus. Phosphate-buffered saline (PBS) and calcium magnesium free (CMF)-PBS, glucose, sodium orthovanadate (Na_3_VO_4_), phenylmethanesulfonyl fluoride (PMSF), aprotinin, protease inhibitor cocktail (IP), Triton X-100, bovine serum albumin, rabbit polyclonal anti-GAPDH antibody (RRID: AB_796208), DNase I, Trypsin, Conduritol B epoxide (CBE), Rock inhibitor, SB431542, SAG, purmorphamine, CHIR99021, cAMP and L-ascorbic acid, mono-dimensional HPTLC were obtained from Sigma-Aldrich (St. Louis, MO, USA). L-Glutamine, penicillin/streptomycin (10,000 Units/mL) and D-MEM were purchased from EuroClone (Paignton, UK). Mouse anti-Neurofilament H (RRID: AB_10694081), rabbit anti-MAP2 (RRID: AB_10693782), mouse anti-Tau (RRID: AB_10695394), rabbit anti-PSD95 (RRID: AB_561221), mouse anti-β3-tubulin (RRID: AB_1904176), rabbit anti-c-Src (RRID: AB_2106059), rabbit anti-phospho-c-Src (p-Src Tyr416) (RRID: AB_331697) and goat-anti-rabbit HRP-conjugated antibodies were obtained from Cell Signalling Technology (Danvers, MA, USA). Mouse anti-TH (RRID: AB_628422) antibody was obtained from Santa Cruz Biotechnology. Mouse anti-LAMP1 (RRID: AB_2296838) antibody was purchased from Developmental Studies Hybridoma Bank (Iowa City, Iowa, USA). The chemiluminescence kit for immunoblotting was obtained from Cyanagen (Bologna, Italy). Neurobasal medium A 1x, Neurobasal medium 1x, B27 Supplement, N2 supplement, KSR supplement, Geltrex, Essential 8 medium, Accutase, 100X non-essential amino acid, 100X 2-mercaptoethanol, 100X Glutamax, EZ-Link Sulfo-NHS-biotin, Dynabeads M-280 Streptavidin and goat-anti-mouse HRP conjugated (RRID: 31430) were obtained from ThermoFisher Scientific (Waltham, MA, USA). Liquid scintillation cocktail Ultima Gold^TM^ was from Perkin Elmer (Waltham, MA, USA). A total of 4–20% Mini-PROTEAN^®^ TGX™ Precast Protein Gels, Turbo Polyvinylidene difluoride (PVDF) Mini-Midi membrane and DC™ protein assay kit were from BioRad (Hercules, CA, USA). High-performance thin-layer chromatography (HPTLC) plate and Triton X-100 were from Merck Millipore (Frankfurten, Germany). LDN-193189 was from Reprocell. FGF-8b, BDNF, GDNF, TGF-3β were from Peprotech (London, UK). 4-Methylumbel-liferyl-b-D-galactopyranoside (MUB-Gal), 4-methylumbelliferyl-b-D-glucopyranoside (MUB-Glc) and 4-methylumbelliferone were obtained from Glycosynth (Warrington, UK).

### 2.2. Generation of Murine Cerebellar Granule Neurons

Pregnant C57BL/6J mice were provided by Charles River—Research Models and Services (Calco, Lecco, Italy). All animal procedures were approved by the Ethics Committee of the University of Milano, Italy and were performed in accordance with the National Institute of Health Guide for the Care and Use of Laboratory Animals (Directive 2010/63/EU) (project numbers FD611.N.TAQ). CGN were prepared as previously described [17]. Briefly, 5-day-old pups were sacrificed by decapitation to extract the cerebellum. CGN were dissociated from pooled cerebella by mechanic trituration with blade (70 times in perpendicular directions), followed by incubation with Trypsin 1% (*w*/*v*) plus DNase I 0.1% (*w*/*v*) in CMF-PBS-Glucose 0.2% (*w*/*v*) (1 mL every 5 cerebella) for 3.5 min at 23 °C. The reaction was stopped by centrifugation of the cell mixture at 1000× *g* for 10 s and the supernatant was removed. The cell pellet was suspended in Trypsin inhibitor 0.04% (*w*/*v*) plus DNase I 0.1% (*w*/*v*) in CMF-PBS-glucose 0.2% (*w*/*v*) (1 mL every 5 cerebella). The cells were definitively dissociated by repeated passages through descending caliber glass Pasteur pipettes. The solution was removed by centrifugation (1000× *g*, 5 min) and the cells were washed with 0.2% (*w*/*v*) glucose in CMF-PBS before being resuspended in Neurobasal A medium containing 25 mM KCl, 1% B27 Supplement, 2 mM L-glutamine, 100 U/mL penicillin, and 100 µg/mL streptomycin. Cells were plated at a density of 3.15 × 10^5^ cells/cm^2^ on plastic supports pre-coated with poly-L-lysine (10 μg/mL for 2 h at 37 °C) and maintained in culture in standard conditions. Then, 24h after plating, the culture medium was supplemented with 0.25 mg/mL Cytarabine. Half of the medium was changed every 2 days.

### 2.3. iPSC Culture

Human iPSC lines derived from fibroblasts of a healthy subject were obtained from the IRCCS Foundation Ca’ Granda Ospedale Maggiore Policlinico. The use of iPSCs was approved by the IRB of Fondazione IRCCS Ca’ Granda Ospedale Maggiore Policlinico Fibroblasts were derived from a 2 mm diameter skin punch biopsy and cultured in DMEM supplemented with 15% fetal bovine serum and 1% penicillin/streptomycin. Fibroblasts were reprogrammed with a commercial kit (CytoTune™-iPS 2.0 Sendai Reprogramming Kit, Thermo Fisher Scientific, Waltham, MA, USA), following the manufacturer’s instructions. Then, 3 weeks after the transfection, colonies were manually picked and individual clones of iPSCs were isolated. The karyotype analysis of selected clones of reprogrammed iPSCs showed no major chromosomal rearrangements (data not shown) and one single clone was further expanded for experiments. Cells showed typical stem cell-like morphology and stained positive for stem cell markers (data not shown). iPSCs were grown in geltrex-coated (1% for 1h at 37 °C) 6-well plates and cultured in Essential 8 Medium. At 80–90% confluence (i.e., every 3–4 days), cells were passaged using Accutase (3 min 37 °C) and plated at a density of 10^4^ cells/cm^2^ in Essential 8 Medium supplemented with 10 µM Rock inhibitor for 24 h.

### 2.4. Differentiation of iPSC into Dopaminergic Neurons

iPSCs were differentiated into DA neurons according to the protocol described by Zhang et al. [18]. Cells were cultured in proper media supplemented with specific factors at proper concentrations as follows. Day 0: KSR differentiation medium (81% DMEM, 15% KSR, 100X 1% non-essential amino acids, 100X 1% 2-mercaptoethanol, 100 U/mL penicillin, and 100 µg/mL streptomycin) supplemented with 10 μM SB431542 and 100 µM LDN-193189. Days 1 and 2: KSR differentiation medium supplemented with 10 μM SB431542, 100 nM LDN-193189, 0.25 μM SAG, 2 μM purmorphamine and 50 ng/mL FGF8b. Days 3 and 4: KSR differentiation medium supplemented with 10 μM SB431542, 100 nM LDN-193189, 0.25 μM SAG, 2 μM purmorphamine, 50 ng/mL FGF8b and 3 μM CHIR99021. Days 5 and 6: 75% KSR differentiation medium and 25% N2 differentiation medium (97% DMEM, 100X 1% N2 supplement, 100 U/mL penicillin, and 100 µg/mL streptomycin) supplemented with 100 nM LDN-193189, 0.25 μM SAG, 2 μM purmorphamine, 50 ng/mL FGF8b and 3 μM CHIR99021. Days 7 and 8: 50% KSR differentiation medium and 50% N2 differentiation medium supplemented with 100 nM LDN-193189 and 3 μM CHIR99021. Days 9 and 10: 25% KSR differentiation medium and 75% N2 differentiation medium supplemented with 100 nM LDN-193189 and 3 μM CHIR99021. Days 11 and 12: B27 differentiation medium (95% Neurobasal medium, 50X 2% B27 supplement, 1% Glutamax, 100X, 100 U/mL penicillin, and 100 µg/mL streptomycin) supplemented with 3 μM CHIR99021, 10 ng/mL BDNF, 10 ng/mL GDNF, 1 ng/mL TGF-b3, 0.2 mM ascorbic acid and 0.1 mM cyclic AMP. From day 13 to the end of differentiation: B27 differentiation medium supplemented with 10 ng/mL BDNF,10 ng/mL GDNF, 1 ng/mL TGF-β3, 0.2 mM ascorbic acid and 0.1 mM cyclic AMP. After 20 days of differentiation, cells were split using Accutase (3 min 37 °C) on geltrex-coated plates at a density of 2 × 10^5^ cells/cm^2^. The medium was changed every day.

### 2.5. Cell Sphingolipid Labelling with [1-^3^H]-Sphingosine

Mature DA neurons at 29 days in culture (DIC) and CGN at 2 DIC were fed with 36 nM [1-^3^H]-sphingosine to metabolically label cell sphingolipids as previously described [19]. Briefly, [1-^3^H]-sphingosine dissolved in methanol was transferred into a sterile glass tube, dried under a nitrogen stream, and then solubilized in an appropriate volume of medium to obtain a final concentration of 36 nM. The correct solubilization was verified by measuring the radioactivity associated with an aliquot of the medium by beta-counter (PerkinElmer, Waltham, MA, USA). After 24 h of incubation the medium was removed and the cells were incubated in a fresh culture medium without radioactive sphingosine.

### 2.6. CBE Treatment

CGN were treated with CBE starting from 3 DIC for 7 and 14 days. Mature DA neurons were treated with CBE starting from 31 DIC for 14 and 29 days. CBE was dissolved in water at a concentration of 100 mM and diluted in culture medium at the final concentration of 0.5 mM [20]. CBE was re-added at every change of the culture medium. Untreated cells were incubated under the same experimental conditions without CBE.

### 2.7. Protein Determination

The protein concentration of samples was assessed with the DC™ protein assay kit according to the manufacturer’s instructions, using BSA at different concentrations as standard.

### 2.8. Cell Surface Protein Biotinylation and Isolation of Plasma Membrane Proteins by Streptavidin Pulldown Assay

At the end of CBE treatment, the control and treated cells were washed with PBS and incubated with 4.7 mg/mL of EZ-Link Sulfo-NHS-biotin in PBS for 30 min at 4 °C. Cells were rinsed with 100 mM glycine in PBS at 4 °C and then mechanically harvested in PBS and centrifuged at 270× *g* for 10 min. Pellets were lysed in 2 mL of 1% Triton X-100, 50 mM Tris-HCl pH 7.4, 150 mM NaCl, 2 mM NaF, 1 mM EDTA, 1 mM EGTA supplemented with Protease Inhibitor Cocktail and 1 mM Na_3_VPO_4_ for 30 min in ice and then passed 11 times in a Dounce homogenizer with tight pestle. The lysates were centrifuged at 1300× *g* for 5 min at 4 °C to remove nuclei and cellular debris and obtain a post nuclear supernatant (PNS). To separate the biotinylated protein fractions, equal amounts of PNS proteins from CBE-treated and untreated cells were mixed with streptavidin-coated magnetic beads (Dynabeads^TM^ M-280 Streptavidin, Thermo Fisher Scientific), stirred overnight at 4 °C, and then recovered by magnet precipitation. Radioactive lipids were extracted from the pellets containing the biotinylated cell surface proteins (P) with chloroform: methanol (2:1, *v*:*v*, 400 µL twice), whereas radioactive lipids from the supernatants containing non-biotinylated proteins (SN) were extracted as described in the lipid analysis paragraph. Radioactivity associated with both fractions was determined by liquid scintillation counting. Lipid separation and quantification were performed as described in the lipid analysis paragraph.

### 2.9. Cell Surface Protein Biotinylation and Isolation of Detergent-Resistant Membrane Fractions

At the end of CBE treatment, the treated and untreated cells were biotinylated as described in the previous paragraph and then mechanically harvested in PBS and centrifuged at 270× *g* for 10 min. Pellets were then lysed in 2 mL of 1% Triton X-100 in 19 mM TNEV buffer (10 mM Tris- HCl, 150 mM NaCl, 5 mM EDTA (pH 7.5)) supplemented with Protease Inhibitor Cocktail and 1mM Na_3_VPO_4_ for 30 min in ice and then lysed 11 times using a tight Dounce. The lysates were centrifuged at 1300× *g* for 5 min at 4 °C to remove nuclei and cellular debris and obtain the PNS. A total of 1 mL of PNS was mixed with an equal volume of 85% sucrose (*w*/*v*) in TNEV buffer containing 1 mM Na_3_VO_4_, loaded at the bottom of a discontinuous sucrose gradient (30–5%) and centrifuged at 200,000× *g* at 4 °C for 17 h. After ultracentrifugation, 12 fractions were collected starting from the top of the tube. The light scattering band, corresponding to the detergent-resistant membrane domain (DRM) fraction, was collected in corresponding fractions 5 and 6, whereas fractions 10, 11 and 12 correspond to high-density fractions (HD). The entire procedure was performed at 0–4 °C in ice immersion. Radioactivity associated with PNS and with gradient fractions (HD and DRM) was determined by liquid scintillation counting [21]. DRM fractions from the CBE-treated and untreated cells were subjected to precipitation of cell surface biotinylated protein and lipids extraction as described in the previous paragraph. In this case, the pellets containing the biotinylated cell surface proteins represent the detergent-resistant plasma membrane domain fractions (PM-DRM). Radioactivity associated with PM-DRM and SN was determined by liquid scintillation counting. Lipid separation and quantification were performed as described in the lipid analysis paragraph. Equal volumes of each fraction were diluted with Laemmli buffer and used for immunoblotting analysis.

### 2.10. Immunoblotting

Immunoblotting for CGN or DA neurons total cell lysates, HD, and DRM fractions were performed using standard protocols. Aliquots of proteins were mixed with Laemmli buffer (0.15 M DTT, 94 mM Tris-HCl pH 6.8, 15% glycerol, 3% *w*/*v* SDS, 0.015% blue bromophenol) and heated for 5 min at 95 °C. Proteins were separated on 4–20% polyacrylamide gradient gels and transferred to PVDF membranes by electroblotting. PVDF membranes were incubated in a blocking solution with (5% non-fat dry milk (*w*/*v*) in TBS-0.1% tween-20 (*v*/*v*)) at 23 °C for 1 h under gentle shaking. Subsequently, PVDF membranes were incubated overnight at 4 °C with primary antibodies diluted in blocking solution. The day after, PVDF membranes were incubated for 1 h at 23 °C with secondary HRP-conjugated antibodies diluted in blocking solution. PVDF were scanned using the chemiluminescence system Alliance Mini HD9 (Uvitec, Cambridge, UK) and band intensity was quantified using ImageJ software (v2.1.0/1.53c) (National Institutes of Health, Bethesda, USA). The following primary antibodies were used for immunoblotting: Polyclonal rabbit anti-PSD95 (dilution: 1:1000), polyclonal rabbit anti-MAP2 (dilution: 1:1000), monoclonal mouse anti-Tau (dilution: 1:1000), monoclonal mouse anti-Neurofilament H (dilution: 1:1000), monoclonal mouse anti-LAMP1 (dilution: 1:100), monoclonal mouse non-phospho-Src Tyr416 (dilution 1:1000), monoclonal mouse phospho-Src family Tyr416 (dilution 1:1000), polyclonal rabbit anti-GAPDH (dilution: 1:7000) and monoclonal mouse anti-β3-tubulin (dilution: 1:1000).

The following secondary antibodies were used: goat-anti-rabbit HRP-conjugated (1:1000) and goat-anti-mouse HRP conjugated (dilution: 1:2000).

### 2.11. Evaluation of Enzymatic Activities in Cell Lysates

GCase activity in total cell lysates was determined as previously described [22,23] using the 4-Methylumbelliferone (MUB)-derived fluorogenic substrate MUB-β-Glc.

Aliquots of cell lysates corresponding to 20 µg of proteins were pre-incubated for 30 min at 23 °C in a 96-well microplate with a reaction mixture composed of: 25 µL of McIlvaine buffer 4X (0.4 M citric acid, 0.8 M Na_2_HPO_4_) pH 5.2, the specific inhibitor of the non-lysosomal β-glucoceramidase AMP-DNM (Adamantane-pentyl-dNM;N-(5-adamantane-1-yl-methoxy-pentyl)-Deoxynojirimycin) at the final concentration of 5 nM and water to a final volume of 75 µL. At the end of pre-incubation, the reaction was started adding 25 µL of MUB-β-Glc at a final concentration of 6 mM. The reaction mixtures were incubated at 37 °C under gentle shaking. At different time points, 10 µL of the reaction mixtures were transferred to a black microplate (Black, 96-well, OptiPlate-96 F, Perkin Elmer) and 190 µL of 0.25 M glycine pH 10.7 were added. The fluorescence associated with MUB was detected by a Victor microplate reader (Perkin Elmer) (ex/em 365/460 nm). Data were expressed as nanomoles of converted substrate/h and normalized to milligrams of cell proteins.

### 2.12. Evaluation of Plasma Membrane Enzymatic Activities in Living Cells

Cells were plated in 96-well microplate at a density of 3.15 × 10^5^ cells/cm^2^ for CGN and 2 × 10^5^ cells/cm^2^ for DA neurons to assay the activity of β-galactosidase and β-hexosaminidase according to a high throughput cell lived-based assay as previously described [22]. After the removal of the culture medium, cells were rinsed with DMEM-F12 without phenol red and artificial substrates (MUB-Gal and MUG), solubilized in DMEM-F12 without phenol red at pH 6 to a final concentration of 1 mM and 6 mM, respectively, which were added to the cells. Reactions occurred at 37 °C for 2 h. At the end of the incubation, 10 µL of the reaction mixtures were transferred to a black microplate (Black, 96-well, OptiPlate-96 F, Perkin Elmer) and 190 µL of 0.25 M glycine pH 10.7 were added. The fluorescence associated with MUB was detected by a Victor microplate reader (Perkin Elmer) (ex/em 365/460 nm). Since substrates are not able to cross the PM, under these experimental conditions, the observed fluorescence is exclusively associated with the PM glycohydrolases. Enzymatic activity was expressed as nanomoles of product/h and normalized to 10^6^ cells.

### 2.13. Lipid Analysis

At the end of CBE treatment, cells were harvested and lysed with H_2_O supplemented with proteinase and phosphatase inhibitors and the cell lysates were subjected to lyophilization. Total lipids were extracted from lyophilized cell lysates with chloroform: methanol: water (2:1:0.1, *v*:*v*:*v*) and separated from the pellet by centrifugation at 13,000× *g* for 15 min, followed by a second and third extraction with chloroform: methanol, 2:1 by vol. Total lipid extracts (TLEs) were subjected to a two-phase partitioning by adding 20% water, resulting in the separation of an aqueous phase (AP) containing gangliosides and an organic phase (OP) containing the other lipids. The radioactivity associated with TLE, AP and OP was determined by liquid scintillation counting by beta-counter (PerkinElmer). Lipids were resolved by mono-dimensional HPTLC using different solvent systems: chloroform: methanol: CaCl_2_ (50:42:11, *v*:*v*:*v*) for the ganglioside analysis of TLE and AP, chloroform: methanol: NH_4_OH (70:30:3, *v*:*v*:*v*) for the glucosylsphingosine analysis of OP and chloroform: methanol: water (110:40:6, *v*:*v*:*v*) for the neutral glycolipid analysis of TLE and OP. Lipids were identified after separation by comigration with authentic standards. Radioactive lipids were detected by radioactivity imaging (Beta-Imager ^T^Racer Betaimager, BioSpace Laboratory, Paris, France) and quantified using the M3 Vision software (BioSpace Laboratory, Paris, France).

### 2.14. Evaluation of GlcCer Release

Cell culture medium was collected daily, centrifuged at 211× *g* to remove cells and filtered through 0.22 µM filtering units. The medium of the CBE-treated and untreated neurons was subjected to dialysis to remove salts and lyophilized and then lipids were extracted. GlcCer released in culture medium was evaluated by HPTLC as described in the lipid analysis section using the solvent system chloroform: methanol: CaCl_2_ (50:42:11, *v*:*v*:*v*).

### 2.15. Proteomic Analysis

All chemicals and reagents used for sample preparation and LC-MS/MS analysis were purchased from Aldrich (Milano, Italy). Digestion buffer was obtained by dissolving NH_4_HCO_3_ in MilliQ water at the final concentration of 50 mM. The reducing solution was 100 mM DTT (dithiothreitol) in digestion buffer, and the alkylating solution was 100 mM IAA (iodoacetamide) in digestion buffer. Trypsin from porcine pancreas was used for protein digestion: the powder was reconstituted in 40 µL of water +0.1% formic acid (FA).

#### 2.15.1. Protein Digestion

A volume corresponding to 50 µg of proteins was transferred to a new Eppendorf tube. Disulfide bonds were reduced by adding 10 µL of reducing solution and incubating samples at 56 °C for 30 min. Cysteine residues were alkylated by adding 30 µL of alkylating solution, vortexing and incubating at RT for 20 min in the dark. After spinning the tubes, proteins were precipitated by adding 1 mL of cold acetone and incubating samples at −20 °C overnight. Samples were then centrifuged at 20,000× *g* for 30 min at 4 °C, the supernatant was removed and the pellet was washed with 100 µL of cold methanol, vortexed for 10 min and centrifuged at 20,000× *g* for 30 min at 4 °C. The supernatant was discarded and protein pellets dried under the fume hood. The pellet was then redissolved in 200 µL of digestion buffer (50 mM NH_4_HCO_3_), and 2 µL of Trypsin (0.5 µg/µL) was added. The samples were incubated at 37 °C in a shaker at 600 rpm overnight for protein digestion. Sample tubes, now containing peptides, were then centrifuged at 20,000× *g* for 30 min at 4 °C. The supernatant was transferred to new Eppendorf tubes and dried down in a speed-vac under vacuum. Peptides were then resuspended in 50 µL of 3% acetonitrile to which was added 0.1% formic acid for LC-MS analysis.

#### 2.15.2. LC-MS/MS Analysis

Tryptic peptides were analyzed by high-resolution LC-MS using a UPLC NanoAcquity chromatographic system (Waters, Milford, MA, USA) coupled with a TripleTof 5600+ mass spectrometer (Sciex, Warrington, UK) equipped with an electrospray ion source. Peptides were desalted using a trapping column (Guard column YMC-Triart C18, 3 µm particle size, 0.5 × 5 mm, 1/32″), then moved on a reversed-phase C18 column (Eksigent C18, 3 µm particle size, 0.3 × 150 mm format) and eluted with a gradient of acetonitrile (ACN) in water. Both eluents were added with 0.1% formic acid. Samples were acquired both in DIA (data-independent acquisition) and DDA (data-dependent acquisition) modes.

#### 2.15.3. Peptide Chromatography

The eluents used were: A (water + 0.1% formic acid) and B (acetronitrile + 0.1% formic acid). Injection volume was 5 µL (Full Loop), the flow rate was set to 5 μL/min, and the column temperature was kept at 45 °C. After trapping the sample on the trap column (at 1% ACN for 5 min), samples were eluted with the following gradient program: 0.0–1.0 min 5% B; 1.0–60.0 min 5 to 40% B; 60.0–63.0 min 40 to 95% B; 63.0–68.0 min 95% B; and 68.0–68.1 min 95 back to 5% B. The column was then reconditioned for 11.9 min. The total run time was 80 min.

#### 2.15.4. SWATH Acquisition

Acquisition in SWATH DIA (data-independent acquisitions), ESI+ was performed. A TOF MS scan was set from 350 to 1250 *m*/*z*. Then, 100 SWATH experiments were collected in the high sensitivity mode, each with an accumulation time of 25 ms, from 100 to 1500 *m*/*z*. The total cycle time was ~2.8 s. Collision energy for each window was automatically calculated by the acquisition software using the equation: CE = 0.063 (*m*/*z*) −3.24 (parameter recommended by the vendor). The ion source parameters were: ion spray voltage floating at 5300 V, ion source gas 1 at 30, curtain gas at 30, and declustering potential at 80 V.

#### 2.15.5. Data-Dependent Acquisition

The spectra were acquired in ESI+. The scan range was set from 300 to 1250 *m*/*z* for MS and from 100 to 1500 *m*/*z* for MS/MS. Precursor ions with charge states 2 and 5 with intensity greater than 150 counts were selected for MS/MS. Collision energy profiles were set according to SCIEX recommended settings. The ion source parameters were: ion spray voltage floating at 5000 V, ion source gas 1 at 20, curtain gas at 30, and declustering potential at 80 V.

### 2.16. Data Analysis

#### 2.16.1. SWATH—Data Analysis

The raw data files were analyzed by using the SWATH Acquisition MicroApp 2.0.1.2133 of PeakView software 2.2 (SCIEX). The quantification was performed applying the following filters: number of peptides per protein at 6, number of transition per peptide at 6, peptide confidence threshold at 99%, FDR threshold at 1%, maximum mass tolerance at 50 ppm, and maximum RT tolerance at 20 min. The modified peptides were excluded. Multivariate data analysis and other statistics were performed by using the free online software MetaboAnalyst [24]. 

#### 2.16.2. DDA—Data Analysis

Raw data were analyzed using Protein Pilot software (SCIEX). The Paragon Method used for protein identification employed a UniProt-reviewed human database; the confidence cut-off was set on 0.05, running the false discovery rate (FDR) analysis. Only protein identified with at least two peptides were retained.

### 2.17. Statistical Analyses

All statistical analyses were performed using GraphPad Prism 7 (GraphPad Software Inc., La Jolla, CA, USA). Data are expressed as mean ± SEM. For normally distributed data, two-tailed unpaired Student’s *t*-test was used. A *p*-value  <  0.05 was considered significant.

## 3. Results

### 3.1. GCase Inhibition Induces a Neurodegenerative Phenotype

We developed two different in vitro neuronal models of GCase deficiency represented by CGN and human iPSCs-derived DA neurons treated with CBE.

The first experimental model is represented by CGN obtained upon the spontaneous differentiation of neuronal precursors isolated from postnatal day 5 C57BL/6 mouse cerebella. After 2 days in culture (DIC), cells were fed with [1-^3^H]-sphingosine in order to label all cell sphingolipids (SLs) at the steady state. When cells reached the stage of CGN (4 DIC), they were treated with 0.5 mM CBE either for 7 days (short-term treatment) or 14 days (long-term treatment) (Appendix A).

The second experimental model exploited the use of human iPSCs (Appendix A) obtained from fibroblasts of a healthy subject and differentiated for 29 days into a neuronal population enriched in DA neurons, according to Zhang P. [18]. As reported in Appendix A, the protocol of neuronal differentiation allows to obtain, with a high yield, cells positive for the neuron-specific class III β-tubulin (Tuj1) and about 35% of cells positive for the DA marker tyrosine hydroxylase (TH). To perform the SL analyses, [1-^3^H]-sphingosine was administered to mature DA neurons (29 DIC). At 31 DIC, cells were treated with 0.5 mM CBE for either 14 days (short-term treatment) or 29 days (long-term treatment) (Appendix A).

Both short- and long-term CBE treatment were able to strongly reduce GCase activity with a residual activity of 4% and 1% in CGN and DA neurons, respectively (Figure 1).

We evaluated the onset of neuronal damage assessing the expression of neuronal-specific markers by immunoblotting analysis both in CGN and in DA neurons after CBE treatment. As shown in Figure 2a, CGN subjected to short-term treatment with CBE did not show any significant variation in the protein levels of the neuronal markers NF-H, MAP2, TAU and PSD95. On the other hand, long-term treated CGN were characterized by a reduction in the protein levels of about 50% for NF-H, 60% for MAP2, 50% for TAU and 30% for PSD95. On the other side, DA neurons showed the onset of neurodegeneration at both time points. Upon short-term treatment with CBE, they were characterized by a reduced expression of about 40% of MAP2 and 70% of TAU (Figure 2b). Upon long-term treatment, the neuronal markers underwent a more evident decrease, with a reduction of 50% for NF-H, 60% for MAP2, 65% for TAU and 35% for PSD95 (Figure 2b). Moreover, we confirmed the degeneration of DA neurons treated with CBE (long-term treatment) by immunofluorescence analyses against MAP2 and Tuj1, showing neuritis fragmentation and degeneration (Appendix A), and by the halving of the cell number compared to the untreated cells (Appendix A).

These data indicate that a long-term inhibition of GCase in CGN and human iPSCs-derived DA neurons resulted in a significant reduction in neuronal markers.

### 3.2. GCase Inhibition Alters the SL Pattern

The radioactive lipids of CGN and DA neurons, untreated or treated with CBE, were analyzed by HPTLC. As shown in Figure 3, both cell models presented a significant and time-dependent accumulation of GlcCer compared to untreated cells. Specifically, short-term CBE-treated CGN and DA neurons showed an increase in GlcCer of seven- and five-fold compared to untreated cells, respectively. The analyses performed after long-term CBE treatment revealed a 10-times higher GlcCer content with respect to untreated cells in both cell models. To validate these data, we analyzed the endogenous counterpart by HPTLC and HPLC elution profile followed by ESI-MS analysis (Appendix A). Moreover, as shown in Figure 3 and Appendix A, upon the accumulation of GlcCer in CBE-treated neurons, we observed an increased production of its deacylated form glucosylsphingosine (GlcSph) in both cell models. The role of GlcSph in the onset of cell toxicity is widely discussed in the literature [25,26,27]; therefore, we decided to investigate its effect on the viability of CGN and DA neurons by its exogenous administration. As shown in Appendix A, no toxic effect was found in CGN upon administration of GlcSph even at high concentrations. DA neurons did not show the onset of cell toxicity at the lower concentration, whereas a 40% decrease of vitality was observed at the higher concentration of GlcSph (Appendix A).

In CGN, we observed a decrease in Cer and SM content, which was more evident upon long-term CBE treatment with a reduction of about 50% and 70%, respectively (Figure 3a). On the other hand, in DA neurons, we did not observe any significant change, except for SM that presented an increase of about 50% upon long-term CBE treatment (Figure 3b).

The analyses of the ganglioside pattern of CGN subjected to long-term CBE treatment revealed a 50% increase in the content of GM3 and GM2 accompanied by a slight reduction in the content of GQ1b (Appendix A). In DA neurons, the effect of long-term CBE treatment on the ganglioside levels was more marked, with an increase of about two-fold in the content of GM3, GM1, and GD1a. Moreover, GD1b and GQ1b exhibited an increase of about 50% and 25%, respectively (Appendix A).

Overall, these results show that, in both cell models, long-term GCase inhibition induced a time-dependent accumulation of GlcCer and GlcSph, together with an increased content of gangliosides.

### 3.3. GCase Inhibition Alters the Proteomic Profile

We performed a label-free expression proteomics analysis on DA neurons subjected or not to long-term CBE treatment. The proteomic analysis identified 70 upregulated (Appendix A) and 71 downregulated (Appendix A) proteins in CBE-treated neurons with respect to untreated ones. The bioinformatics analyses by DAVID software [28] (Appendix A) showed that, among the upregulated proteins, there was a significant enrichment in proteins involved in the lysosomal structure and in the glycosphingolipids catabolism. In addition, we observed also an enrichment in proteins related to mitochondria activity and glycoproteins biosynthesis and trafficking (Figure 4a). Among the significantly enriched pathways, we further refined the investigation on the “sphingolipid metabolism” highlighted by DAVID. By using STRING, a database for interaction analysis and gene enrichment [29], we found that galactosylceramide catabolism is significantly enhanced following inhibition of GCase, mainly through the overexpression of galactosylcerebrosidase and prosaposin (Appendix A). This result comes with little surprise, with the cell reacting to GCase inhibition by activating alternative sources for ceramide production.

On the other hand, the downregulated proteins were mainly integral component of membranes and trafficking proteins, such as RAB5, which suggested an impairment in the endocytosis processes, and RAB6, which is involved in the ER-Golgi-PM vesicular trafficking (Figure 4b). These data indicate that GlcCer accumulation due to GCase inhibition strongly affects neuronal proteostasis.

### 3.4. GlcCer Accumulation Affects the Lysosomal Compartment

A widely described mechanism occurring upon lysosomal impairment is the nuclear translocation of the transcription factor EB (TFEB), which is a master regulator of the expression of genes coding for proteins involved in compensatory processes aimed at counteracting the lysosomal accumulation of uncatabolized substrates [30,31]. We found that the nuclear translocation of TFEB was enhanced in both cell models after long-term CBE treatment (Appendix A). One of the described consequences of TFEB transcription activity is the activation of lysosomal biogenesis [30]. We evaluated the protein levels of the lysosomal marker LAMP1 in both cell models after long-term CBE treatment. As shown in Figure 5, LAMP1 expression was found to be increased of about 30% and 50% in CBE-treated CGN and DA neurons, respectively.

The staining of CGN with Lysotracker Red DND-99 indicated that CBE treatment caused an increase in the relative volume of the endo-lysosomal compartment (Appendix A). The indirect immunofluorescence staining of LAMP1 in DA neurons corroborated the increased expression of the lysosomal marker after CBE treatment, suggesting an augmented number of lysosomes (Appendix A). Electron microscopy analysis of DA neurons subjected to long-term CBE treatment confirmed the increase in the number of lysosomes and also suggested an increase in their size (Appendix A). Furthermore, we investigated the lysosomal catabolic potential by feeding untreated and long-term CBE-treated CGN with radioactive [3-^3^H(sphingosine)]GM1 [32]. The use of GM1 derivative, which is radioactive on the sphingosine moiety, allowed to follow the production of its catabolites in living cells, giving an indication of lysosomal catabolic activity. The uptake of the radioactive ganglioside was the same in both untreated and CBE-treated CGN (data not shown). As shown in Appendix A, we found a reduced production of radioactive GM1 catabolites, such as GM3, lactosylceramide, GlcCer and Cer in CBE-treated CGN, suggesting an impairment of lysosomal catabolism. Of note, the reduction in radioactive globoside Gb3, whose biosynthesis depends on 3-^3^H(sphingosine) salvage pathway, further supported lysosomal dysfunction in CBE-treated neurons.

Taken together, these results indicate that long-term GCase inhibition induced the biogenesis of lysosomes, which in turn presented a reduced catabolic potential, suggesting an impairment of lysosomal activity.

### 3.5. GCase-Inhibited Neurons Are Characterized by Aberrant Lysosomal Exocytosis

It is already well established that lysosomal impairment and TFEB transcription activity lead to the activation of aberrant lysosomal exocytosis [30]. The fusion of lysosomes with the PM induces the release of lysosomal content in the extracellular environment and the association of lysosomal glycohydrolases with the external leaflet of the PM. To investigate this process, the enzymatic activity of specific lysosomal glycohydrolases was evaluated at PM level [22]. As shown in Figure 6a, in long-term CBE-treated CGN we found an increase of about 50% in the activity of the PM-associated β-galactosidase. In long-term CBE-treated DA neurons, there was an increase of about two-fold in the activity of both the PM-associated β-galactosidase and β-hexosaminidase (Figure 6b).

Subsequently, we observed an increase in the release of GlcCer in the extracellular milieu of about 50% in CBE-treated CGN and 25% in DA neurons compared to untreated cells (lower panel Figure 6a,b).

These data demonstrate that long-term GCase inhibition in CGN and DA neurons caused an increased lysosomal exocytosis and consequently led to the release of the uncatabolized GlcCer in the extracellular milieu.

### 3.6. GlcCer Accumulation Is Not Confined to Lysosomes

In other LSDs, the impairment of the lysosomal compartment due to the accumulation of uncatabolized substrates is accompanied by modifications in the lipid composition of the PM [30]. To investigate this aspect, we fed CGN and DA neurons with [1-^3^H]-sphingosine that was equally incorporated by neurons (data not shown).

At the end of long-term CBE treatment, PM proteins and the surrounding lipids were isolated exploiting a biotin–streptavidin affinity pull-down assay. The radioactive lipids were extracted from the precipitate (P) and from the supernatant (SN) obtained after precipitation and the corresponding proteins were denatured to assess the efficiency of precipitation. The biotinylated proteins recovered in the P fraction were about 80% in CGN and 90% in DA neurons independently from the CBE treatment (data not shown). In DA neurons, precipitated proteins were then identified by high resolution mass spectrometry, following digestion with trypsin. A list of 72 proteins was confidently identified as exclusively expressed at the PM level of CBE treated neurons (Appendix A). An interaction analysis reveals that 44 out of 72 proteins show significant functional association, as demonstrated in Appendix A. Furthermore, several molecular functions were found to be significantly enriched in this organic insoluble fraction (Appendix A). We then explored possible known functional associations between this dataset and the target GCase. No major molecular interactions were highlighted by our analysis.

Regarding the radioactive lipids, 41% was associated with the P fraction of untreated CGN, while 61% was recovered in the P fraction obtained from CBE-treated CGN. Similarly, in the P of untreated DA neurons, we found 40% of radioactivity which increased up to 60% in case of CBE treatment (data not shown). As shown in Figure 7, the analyses of the radioactive lipids of the SN and P revealed that in CBE-treated CGN and DA neurons, the accumulation of GlcCer in the SN was about 12-fold and 6-fold compared to untreated neurons, respectively. Interestingly, a more marked increase was observed in the P fraction, which was approximately 30-fold and 18-fold higher in CBE-treated CGN and DA neurons, respectively (Figure 7).

These data show that the chronic inhibition of GCase induced changes in the PM protein composition and accumulation of GlcCer.

### 3.7. GCase Inhibition Induces Changes in PM Microdomains

In the PM, SLs together with cholesterol and a selected number of proteins organize macromolecular complexes involved in the control of signaling cascades [33,34]. Based on these considerations, we analyzed these microdomains (DRM) in DA neurons treated or not with CBE [33].

To this purpose, DRM domains were isolated from DA neurons subjected to long-term CBE-treatment. DRM represented the low-density detergent insoluble fractions (fraction 5 and 6) obtained by loading the same protein amount of cell lysates on a sucrose density gradient, while the high-density fractions (HD) contained all the remaining cell components. The 12 collected fractions of both CBE-treated and -untreated DA neurons were assessed for the presence of flotillin, a DRM marker, and calnexin, a typical non-DRM-associated protein, as well as for the radioactivity content. As shown in Appendix A, flotillin was enriched in fractions 5 and 6, whereas calnexin was detectable only in HD fractions 10, 11 and 12.

The radioactivity associated with SLs was enriched in fractions 5 and 6 with respect to the fractions 10, 11 and 12 (Appendix A). Based on these data, fractions 5 and 6 were pooled together as DRM, while fractions 10, 11 and 12 were considered as HD.

Interestingly, the radioactivity associated with DRM of untreated DA neurons was about 41% of the total and increased up to 66% in CBE-treated cells (Appendix A). The analyses of the radioactive lipids revealed that DRM fractions were enriched in SLs at the expense of phosphatidylethanolamine (PE) when compared to HD (Appendix A).

From the SL pattern of DRM, it clearly emerged that in CBE-treated DA neurons the increased radioactivity was mainly due to an augmented content of GlcCer (Figure 8a). To evaluate the effect of CBE only in PM-DRM excluding any contribute of DRM from intracellular membranes, we submitted DRM to biotin–streptavidine affinity pull-down assay in conditions preserving DRM integrity. The radioactive lipids were extracted from the P and SN fractions obtained after precipitation, and the corresponding proteins were denatured to assess the efficiency of precipitation. About 90% of the biotinylated proteins were recovered in the P, independently from the CBE treatment (data not shown). In addition, the P fraction of untreated and CBE-treated neurons contained, respectively, 95% and 90% of the total amount of radioactive lipids associated with DRM (data not shown).

The analyses of the radioactive lipid pattern showed that PM-DRM of CBE-treated DA neurons were characterized by a 11-fold increase in GlcCer content and by the halving of Cer levels compared to untreated DA neurons (Figure 8b). Conversely, a two-fold enrichment in the active form of c-Src (pTyr416-Src) was found in DRM of CBE-treated DA neurons with respect to untreated cells (Figure 8c).

Overall, these results show that GlcCer accumulated in specific PM signaling microdomains, altering their SL and protein pattern.

## 4. Discussion

Mutations in the *GBA1* gene are associated with several neurodegenerative disorders, such as Gaucher disease, Parkinson’s disease, dementia with Lewy bodies, rapid eye movement disorder, and sleep behavior disorders [1,35,36,37,38,39]. Nevertheless, the pathogenic role of mutated GBA1 is still a cause of debate in the scientific community. An increasing amount of evidence supports GCase loss of function and the consequent impairment of GlcCer metabolism at the basis of the onset of neurodegenerative phenotypes [13,40]. The main limitation of the study of the molecular mechanisms linking GlcCer accumulation to the onset of neuronal damage is the lack of suitable in vitro or in vivo models. Indeed, studies on post-mortem human brain derived from GD patients are limited by their low availability, rapid deterioration, and impossibility to perform analyses on the involved molecular pathways. Moreover, *GBA1* knock-out mice with complete enzyme deficiency die within 24 h after birth, while single-allele knockout mice do not show nigrostriatal degeneration or PD-like phenotype. Finally, immortalized cells carry intrinsic artefacts, undergoing continuous mutations during the different passages in culture, thus presenting an unstable genotype and consequently a variable phenotype. In addition, patient-derived iPSCs are influenced by the donor patient’s genotype [41,42,43].

Here, we proposed to develop novel in vitro models able to recapitulate, in a reasonable timing, the phenotype of neurons affected by GCase deficiency. In particular, we took advantage of two different neuronal cultures: one represented by CGN derived from pups of WT C57BL/6 mice and the other consisting of human DA neurons differentiated from iPSCs derived from a healthy subject. Both neuronal models were treated with CBE to induce GCase deficiency [16]. We selected two different neuronal populations derived from distinct species to test whether the effect of GCase inhibition on neuronal homeostasis was shared by neurons with different activity and origin.

Interestingly, we discovered that long-term GCase inhibition is required to induce a sufficient GlcCer accumulation able to cause a neurodegenerative phenotype, suggesting the existence of a tolerance threshold that, once overcome, leads to cell toxicity. In GD patients, the lysosomal accumulation of GlcCer causes the aberrant production of GlcSph by the action of the acid ceramidase [44]. Of note, considering the detergent-like physical-chemical properties of GlcSph, a possible role of this catabolite in the onset of neurodegeneration has been hypothesized [25,27]. To exclude its involvement in the onset of neuronal damage in our model, we verified that GlcSph does not exert any toxic effect at the concentration found in GD patients, whereas the treatment with high concentration of GlcSph partially affects the viability of DA neurons only. In addition, we can exclude the toxic effect of another sphingolipid known to be involved in the activation of the apoptotic process, the ceramide. In particular, we found that its level decreased in CGN, probably because cells do not activate a compensatory mechanism involved in its de novo biosynthesis in order to deal with the reduced availability of sphingosine due to the block of glucosylceramide catabolism. Conversely, we can speculate that such mechanism was activated in DA neurons, since the levels of ceramide are the same in treated and untreated neurons.

Interestingly, label-free expression proteomic analysis identified changes in the expression of 141 proteins in CBE treated DA neurons with respect to untreated cells. In particular, the expression of 70 proteins was upregulated, while 71 proteins were downregulated, suggesting that the GlcCer accumulation due to GCase inhibition has a general impact on neuronal proteostasis.

Our models also recapitulate the secondary accumulation of uncatabolized complex SLs, such as gangliosides, which is a common feature observed in several LSDs due to an impairment in the catabolic activity of the endo-lysosomal compartment [45,46]. Nowadays, we know that cells can counteract this impairment by activating a compensatory mechanism, which includes the activation of the transcriptional factor EB, responsible for the promotion of lysosomal biogenesis and exocytosis. These two evolutionary conserved events are activated to reduce as much as possible the accumulated material and restore the proper lysosomal function [29,30,46]. In particular, the release of material in the extracellular milieu could be attributed to cells belonging to organs characterized by an intense vascularization or by a parenchyma formed by cells that can recycle or store the released material. Notably, we demonstrated that the accumulation of uncatabolized substrate induces the promotion of lysosomal exocytosis. This observation suggests a new pathogenic mechanism, since in patient’s brain the material released by impaired neurons could be poorly absorbed by the endothelial cells and could activate astrocytes and glial cells, turning on neuronal inflammation and allowing the self-propagation of the disease to other brain areas [47].

In addition, we found that, in GCase-deficient neurons, which accumulate high levels of GlcCer, the aberrant activation of lysosomal exocytosis is also responsible for alterations in the PM architecture. Indeed, we found that GlcCer accumulation is not only confined to lysosomes but occurs also at PM level, affecting also its protein composition. In particular, by high resolution mass spectrometry we found a list of 72 proteins that were only expressed at the PM of CBE treated neurons.

GSLs are particularly enriched in membrane domains called “lipid rafts” or “DRM” [48,49,50]. These discontinuous portions of the PM are fundamental signaling platforms through which cells can regulate extracellular stimuli and modulate the signal transduction [51,52]. Structural alterations have been described in DRM derived from organs of GD animals and from in vitro GD models represented by immortalized cell lines [53,54]. Conversely, no data exist on the effect of GCase deficiency in DRM of neurons.

Based on these premises, we focused our analysis on PM DRM obtained from CBE-treated and untreated DA neurons, where we observed the accumulation of GlcCer. In these domains, SLs spontaneously segregate, determining their enrichment at the expense of glycerophospholipids. For this reason, the GlcCer inserted in the PM, preferentially localizes in these specific areas. This could explain the different GlcCer enrichment observed by the analysis of the entire PM or PM DRM.

An increasing amount of evidence has shown the importance of DRM in the maintenance of cell homeostasis and modifications in their structure are often associated with important changes in cell functioning and phenotype [34,55]. Remarkably, we demonstrated for the first time that, in pathological neurons, the accumulation of GlcCer occurs also in the PM DRM, causing alterations of the protein composition of these domains. In particular, we found a local hyperactivation of c-Src, a non-receptor protein kinase involved in several intracellular signaling pathways, fundamental for the neuronal homeostasis [56,57].

Usually, the activation of c-Src induces its own translocation outside the DRM, event necessary to exert its functions. In particular, the translocation of the active form of c-Src is promoted by the enrichment of ceramide in DRM [58,59]. In our models, the reduction in ceramide due to the accumulation of GlcCer within PM DRM could sequester the activated c-Src, avoiding its functioning.

Even though this aspect needs to be further investigated, it opens a new perspective on putative targets for novel therapeutic strategies.

Taken together, our data indicate a direct role of the accumulated GlcCer in the onset of neuronal damage (Appendix A). Interestingly, we observed GlcCer accumulation also at the PM level, in particular in DRM domains. In addition, our observations strengthened the existence of a pathogenic lysosome–PM axis that determines alterations of the PM architecture, contributing to the establishment of the neurodegenerative phenotype. Our work establishes two reliable neuronal models to study the impact of GCase deficiency in neurons and opens a new scenario on the study of GCase-related neurodegenerative disorders.

## Figures and Tables

**Figure 1 cells-11-02343-f001:**
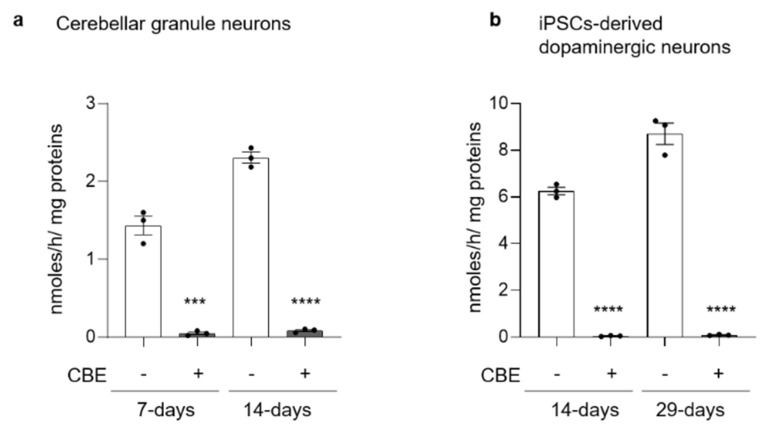
Effect of conduritol B epoxide treatment on β-glucocerebrosidase activity. β-glucocerebrosidase activity was evaluated in (**a**) mouse cerebellar granule neurons treated or not with 0.5 mM CBE for 7 or 14 days and (**b**) in human iPSC-derived dopaminergic neurons treated or not with 0.5 mM CBE for 14 or 29 days. β-glucocerebrosidase activity is expressed as nmoles of formed product/hour/mg cell proteins. All data are shown as mean ± SEM of three different experiments. *** *p* < 0.001, **** *p* < 0.0001, two-tailed Student’s *t*-test vs. CBE-untreated cells.

**Figure 2 cells-11-02343-f002:**
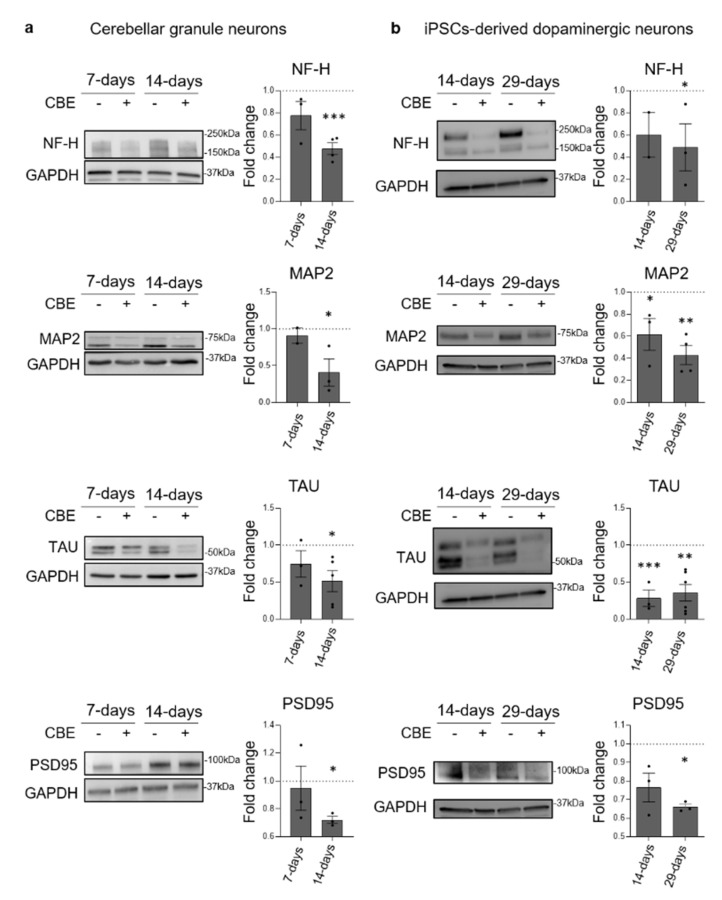
Effect of β-glucocerebrosidase inhibition on neuronal markers’ expression. Immunoblotting analysis of the expression of Neurofilament H (NF-H), MAP2, TAU and PSD95 in: (**a**) mouse cerebellar granule neurons (CGN) treated or not with conduritol B epoxide (CBE) 0.5 mM for 7 and 14 days and (**b**) human iPSC-derived dopaminergic neurons treated or not with CBE 0.5 mM for 14 and 29 days. Optical densities of the individual bands were quantified using NIH ImageJ and normalized to GAPDH. Data are expressed as fold change with respect to CBE-untreated cells (dashed lined) and are the mean ± SEM of three different experiments. * *p* < 0.05; ** *p* < 0.01; *** *p* < 0.001. Two-tailed Student’s *t*-test vs. CBE-untreated cells.

**Figure 3 cells-11-02343-f003:**
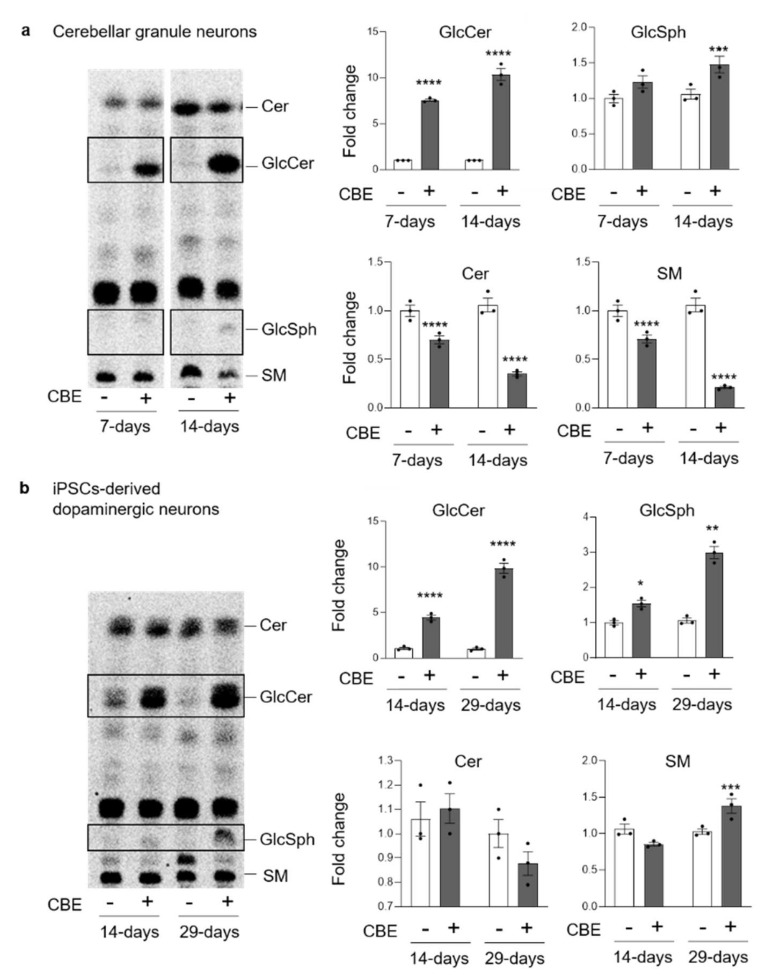
Effect of β-glucocerebrosidase inhibition on glucosylceramide and glucosylsphingosine levels. Representative digital autoradiography of sphingolipid pattern and quantification of the radioactivity associated with glucosylceramide (GlcCer), glucosylsphingosine (GlcSph), ceramide (Cer) and sphingomyelin (SM) in: (**a**) mouse cerebellar granule neurons fed with radioactive sphingosine and treated or not (−) with 0.5 mM conduritol B epoxide (CBE) for 7 and 14 days and (**b**) human iPSC-derived dopaminergic neurons fed with radioactive sphingosine and treated or not with CBE for 14 and 29 days. Data are expressed as fold change with respect to CBE-untreated cells (−) and are the mean ± SEM of three different experiments. * *p* < 0.05; ** *p* < 0.01; *** *p* < 0.001; **** *p* < 0.0001. Two-tailed Student’s *t*-test vs. CBE-untreated cells.

**Figure 4 cells-11-02343-f004:**
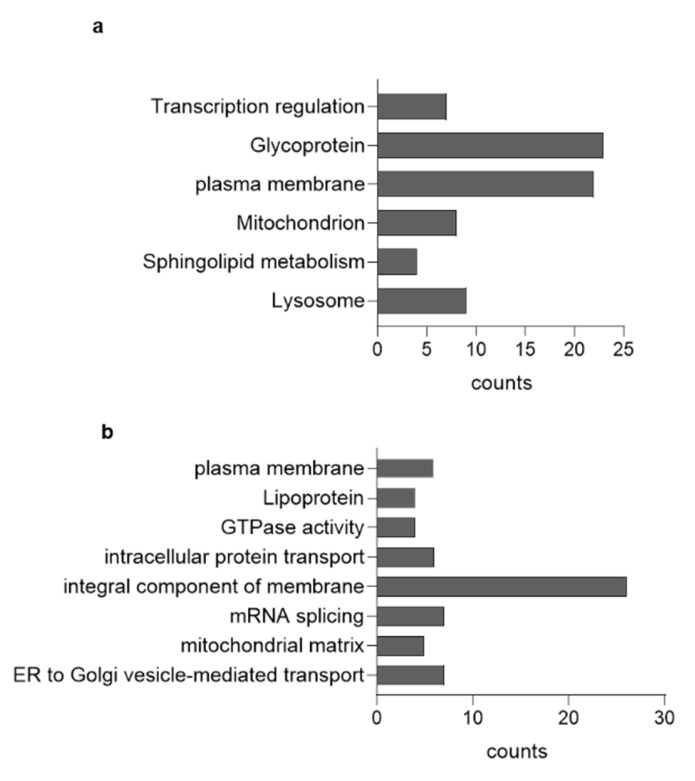
Effect of β-glucocerebrosidase inhibition on proteomic profile. Bar chart showing the cluster enriched annotation groups of proteins (**a**) upregulated and (**b**) downregulated in CBE-treated DA neurons with respect to untreated cells obtained by David bioinformatics analysis. The x-axis indicates the number of proteins involved in each term.

**Figure 5 cells-11-02343-f005:**
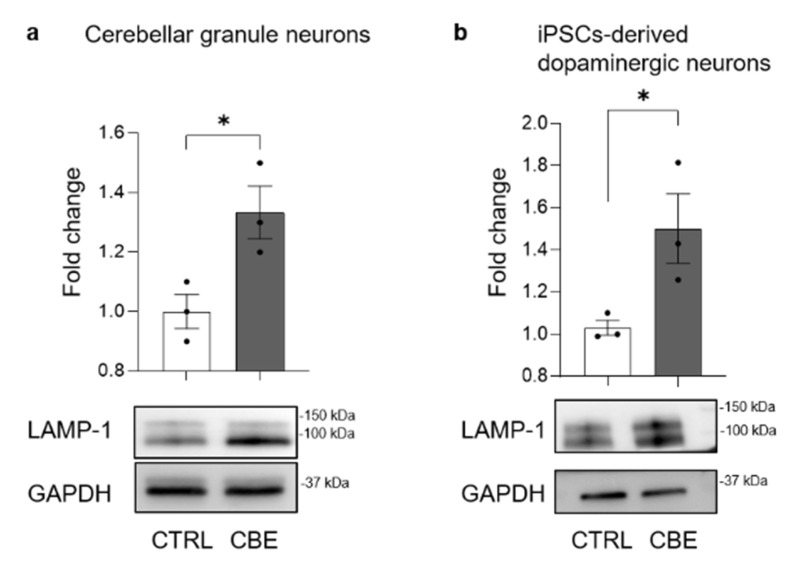
Effect of glucosylceramide accumulation on lysosomal-associated membrane protein-1 expression. Immunoblotting analyses of lysosomal-associated membrane protein (LAMP-1) in (**a**) mouse cerebellar granule neurons and (**b**) human iPSC-derived dopaminergic neurons treated or not with 0.5 mM conduritol B epoxide (CBE) for 14 and 29 days, respectively. Data are expressed as fold change with respect to CBE-untreated cells (CTRL) and are the mean ± SEM of three experiments; * *p* < 0.05, two-tailed Student’s *t*-test vs. CBE-untreated cells.

**Figure 6 cells-11-02343-f006:**
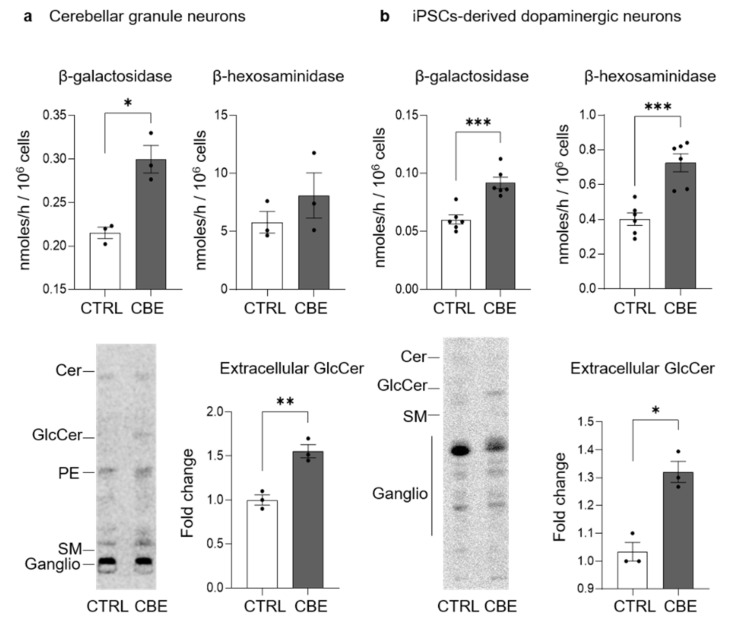
Activity of plasma membrane glycohydrolases and extracellular release of glucosylceramide upon its accumulation. Specific enzyme activity of plasma membrane-associated β-galactosidase and β-hexosaminidase and representative image and quantification of extracellular release of glucosylceramide (GlcCer) in (**a**) mouse cerebellar granule neurons and (**b**) human iPSC-derived dopaminergic neurons fed with radioactive sphingosine followed by treatment in presence or in absence of 0.5 mM conduritol B epoxide (CBE) for 14 and 29 days, respectively. Enzyme activities are expressed as nmoles/ h normalized to 10^6^ cells and are the mean ± SEM of three experiments (* *p* < 0.05, *** *p* < 0.001, two-tailed Student’s *t*-test vs. CBE-untreated cells (CTRL)). Extracellular release of radioactive GlcCer was determined after the lipid extraction of cell culture medium and HPTLC separation (Cer = ceramide, PE = phosphatidylethanolamine, SM = sphingomyelin, ganglio = gangliosides). Data are expressed as fold change with respect to CBE-untreated cells (CTRL) ad are the mean ± SEM of three experiments; * *p* < 0.05, ** *p* < 0.01, *** *p* < 0.001, two-tailed Student’s *t*-test vs. CBE-untreated cells.

**Figure 7 cells-11-02343-f007:**
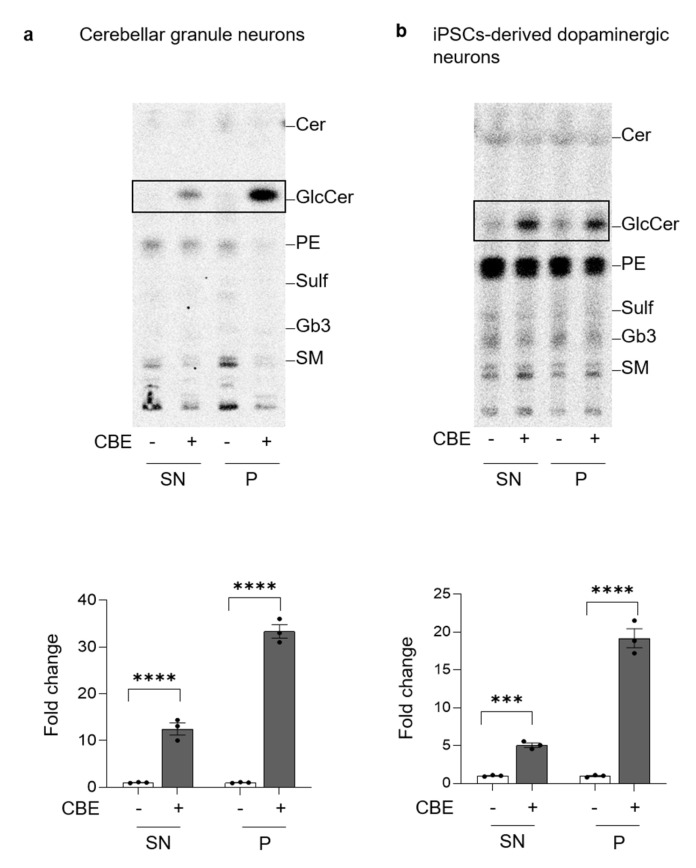
Accumulation of glucosylceramide at the cell surface. Representative HPTLC image of radioactive lipids of cell fractions obtained after selective cell surface protein biotinylation followed by magnetic separation using Dynabeads™ streptavidin magnetic beads; lipids were extracted from the supernatants containing non biotinylated proteins (SN) and the pellets containing the biotinylated cell surface proteins (P); (**a**) mouse cerebellar granule neurons and (**b**) human iPSC-derived dopaminergic neurons treated or not (−) with 0.5 mM conduritol B epoxide (CBE), for 14 and 29 days, respectively. (Cer = ceramide; GlcCer = glucosylceramide; PE = phosphatidylethanolammine; SM = sphingomyelin). Data are expressed as fold change of GlcCer with respect to CBE-untreated cell fraction (−) and are the mean ± SEM of three experiments (*** *p* < 0.001; **** *p* < 0.0001; one-way two-tailed Student’s *t*-test vs. CBE-untreated cells). GlcCer content was normalized nCi/ mg of proteins of the sample.

**Figure 8 cells-11-02343-f008:**
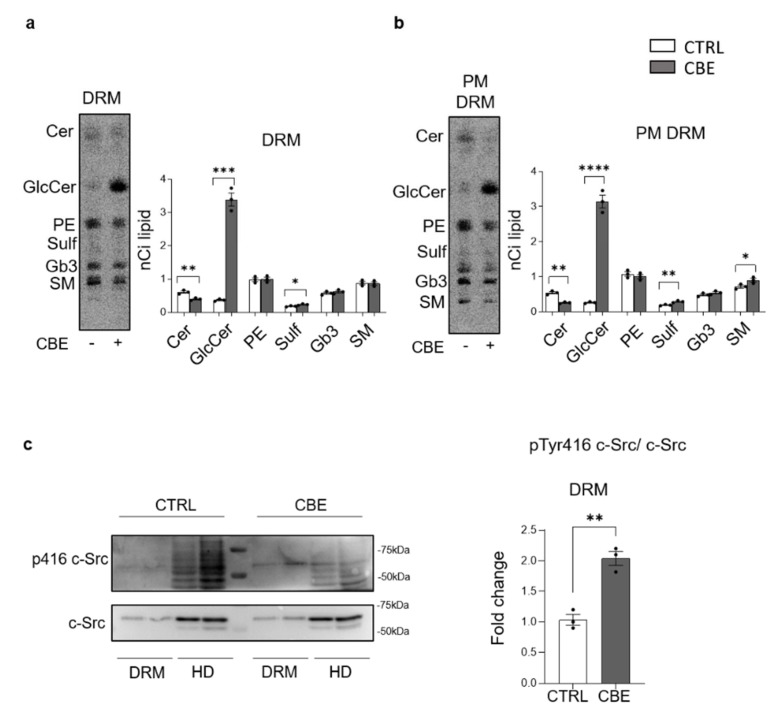
p416 c-Src expression and lipid distribution in detergent-resistant plasma membrane fractions of iPSCs-derived dopaminergic neurons treated or not with conduritol B epoxide. (**a**) Representative image of HPTLC of radioactive lipids of detergent-resistant membrane domain fraction (DRM) of human iPSCs-derived dopaminergic neurons treated or not (−) with 0.5 mM conduritol B epoxide (CBE) for 29 days. (**b**) Representative image of HPTLC of radioactive lipids of detergent-resistant membrane domain fraction (DRM) after cell surface biotinylation of human iPSCs-derived dopaminergic neurons treated or not (−) with 0.5 mM conduritol B epoxide (CBE) for 29 days. Detergent-resistant plasma membrane domain fraction (PM-DRM) was obtained from DRM by streptavidine magnetic beads separation. Data are expressed as nCi and are the mean ± SEM from at least three experiments (* *p* < 0.05; ** *p* < 0.01; *** *p* < 0.001; **** *p* < 0.0001; two-tailed Student’s *t*-test). (Cer = ceramide; GlcCer = glucosylceramide; PE = phosphatidylethanolamine; Sulf = sulphatides; Gb3 = globo-triaosylceramide; SM = sphingomyelin). (**c**) Immunoblotting analysis of Src and pTyr416 c-Src expression in DRM fraction and non-DRM high density fractions (HD) of dopaminergic neurons treated or not with CBE for 29 days; analysis of pTyr416 c-Src activation: pTyr416 c-Src band intensity was normalized to c-Src signal of the same sample. Data are expressed as fold change with respect to CBE-untreated cell fraction of and are the mean ± SEM of at least three experiments; ** *p* < 0.01; two-tailed Student’s *t*-test.

## Data Availability

The data sets generated and analyzed during the current study are available from the corresponding author on reasonable request.

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
