# Peer review of "β-Glucocerebrosidase Deficiency Activates an Aberrant Lysosome-Plasma Membrane Axis Responsible for the Onset of Neurodegeneration"

_cells, 2022, doi:10.3390/cells11152343_

Round 1
Reviewer 1 Report
Comments to Authors:
Lunghi and Co-workers presented good work in the manuscript entitled “β-glucocerebrosidase deficiency activates an aberrant lysosome-plasma membrane axis responsible for the onset of neurodegeneration". This seems to be an interesting work where they have tried to explore the molecular mechanism involved in the onset of neurodegeneration linking to the deficiency in β-glucocerebrosidase activity. The two different in vitro neuronal models of GCase deficiency by an irreversible β-glucocerebrosidase inhibitor (conduritol B epoxide) were studied in cerebellar granule neurons (CGN) and human iPSCs-derived DA neurons. Overall, a lot of experiments were conducted. They have provided lots of data to prove the GlcCer accumulation role at the lysosome - Plasma membrane PM) axis in Gaucher’s disease. They finally concluded that β-glucocerebrosidase function loss impairs the lysosomal compartment, establishing the lysosome-PM axis responsible for modifications in the plasma membrane architecture and probable alterations of intracellular signaling pathways, leading cause of neuronal damage. For improving the manuscript, the following comments should be addressed.
The comments to address by the authors:
-. There are some typographical errors in the Manuscript, please correct them such as H20 to H2O. Please check for grammatical mistakes also and correct them seriously.
-. Please prepare an abbreviation list that should collectively consist of all forms of abbreviation mentioned throughout the manuscript
-. Why did the Authors have inhibited the specific β-glucocerebrosidase activity and not used the genetic knockout cells and animals in their experiment? They could use genetic models knockout β-glucocerebrosidase CGN and dopaminergic cells to characterize the role of β-glucocerebrosidase, please justify??
-. What was the reason behind using 7, 14 days treatment of CBE in CGN cells and 14, 29 days CBE treatment in iPSC-derived dopaminergic neurons? If so then why did the authors not perform 14, 29 days treatment of CBE in CGN cells and 7, 14 days treatment of CBE in iPSC-derived dopaminergic neurons?
-. It would be good if the authors provide a graphical abstract that will draw a great overview of their complete hypothesis?
-. Why the authors did not perform toxicity studies of GlcSph in iPSC dopaminergic neurons (Fig. S7)?
-. Fig. S8a, Ganglioside pattern experiment; CBE treatment the GM3, GQ1b, GD1a in CBE treatment the appearing bands are not cleared, The reviewer would like to suggest authors repeat this experiment? Similarly, repeat the experiment for DA neurons.
-. Appreciate the results of LAMP1 western blot, immunofluorescence (DA neurons), and Electron microscopy. Please repeat the western blot experiment for LAMP1 in both CGN and DA neurons? It would be good if the author also please check the p62 and LC3-II/I expression levels in both the CGN and DA neurons with CBE treatment?
-. It would be more interesting if the authors check the alpha-synuclein accumulation propensity in β-glucocerebrosidase inhibited CGN and DA neurons?
-. Please improve the discussion part and check references must be as per journal standard.
-. Please avoid irrelevant self-citations if not relevant to study and check plagiarism before submission.
-. In Fig. 1, Harsh condition, the degree of GBA1 deficiency in PD was 30-40% deficiency of GBA1 enzyme activity.
-. Page 2, line 54, substantia nigra pars compactaïƒ not italicized
-. Page 2, line 54, PMID: 34644545 should be cited for the finding.
-. Page 2, lines 65-67, PMID: 29311330 should be cited for the finding.
-. Page 2, lines 69-71, the in vitro models of GCase deficiency using CBE treatment in murine cerebellar granule neurons and iPSCs-derived DA neurons do not represent GBA1-associated PD.
-. Page 3, line 116, the reason for the use of murine cerebellar granule neurons is unclear. The neurons are not affected in GBA1-associated PD.
-. Page 3, line 139, No information for iPSCs. No information for review and approval from Institutional Stem Cell Research Oversight Committee to use human stem cell.
-. Page 3 and 4, more characterization data are required for iPSCs and DA neurons and should be presented. Characterization of iPSC-1) Immunocytochemical and Western blot analysis of the expression of endogenous pluripotent markers; AP, SSEA4, NANOG, TRA-1-60, OCT4, and SSEA3, 2) Karyotypes, 3) If possible Teratoma assay.
Characterization of DA neurons-1) immunocytochemical and Western blot analysis of the expression of the mature neuronal markers; TH, TUJ1, MAP2, PITX3, Nurr1, and FOXA2, 2) Electrophysiological properties such as DA levels from HPLC, Spontaneous firing and action potentials, voltage-gated K+ and Na+ currents
-. The reviewer encourages you to use GBA1 as a nomenclature.
-. Changes in mRNA should be informative.
-. In Sup fig. 2, Quantification of TH positive neurons is required.
-. Page 6, line 271, GCase activity should be also measured in lysosomal fraction.
-. In Fig. 1, 4% GBA1 enzyme activity is detected, Why is that? Is that GBA2 enzyme activity?
-. In Fig. 2, the reviewer is not clear why the author conducted the experiments (move to supplementary data).
-. Page 8, line 412, the subtitle is overstated and Page 10, lines 466-468, the conclusion is also overstated. The presented data do not represent the neurodegenerative phenotype observed in GCase-related pathologies.
-. Page 10, lines 483-484, the sentence should be rewritten.
-. Page 10, lines 487-488, the authors should discuss why the levels of Ceremide were not changed in the DA neurons in discussion.
-. In Sup Fig. 8. the authors should discuss why there is a difference between granule neurons and DA neurons.
-. In Sup fig. 8, standard maker for HPTLC is missing.
-. In Fig 4, validation study is required in a few proteins identified from Masspec.
-. In Fig 5, The reviewer does not see change in Lamp1 in both cells.
-. Please use a dot plot in all data.
-. In Sup Fig. 9C, should be quantified for lysosome biogenesis (number and size). TFEB levels would be informative.
-. Page 14, lines 603-604, HPTLC image for extracellular GlcCer should be presented.
-. In Fig. 7 Standard marker is required for the TLC.
-. Page 16, line 655, check the references.
-. Page 17, lines 724-725, it looks wrong. GlcCer is accumulated in GBA1-PD iPSC derived DA neurons (PMID: 29311330).
-. Fig. 8 Standard marker is required for the TLC.
-. What is the syn levels, Is alpha-synuclein oligomer observed? Does CBE increase the half-life of alpha-synuclein ?
Author Response
Reviewer #1:
We thank the reviewer for the comments that we addressed as detailed below.
Comment: There are some typographical errors in the Manuscript, please correct them such as H20 to H2O. Please check for grammatical mistakes also and correct them seriously.
Answer: We checked the manuscript for the typos and grammatical mistakes. All the changes are tracked.
Comment: Please prepare an abbreviation list that should collectively consist of all forms of abbreviation mentioned throughout the manuscript.
Answer: We addressed the request of the reviewer and we prepared an abbreviation list that we added at the end of the manuscript but we do not know if it address the editorial request of the journal.
Comment: Why did the Authors have inhibited the specific β-glucocerebrosidase activity and not used the genetic knockout cells and animals in their experiment? They could use genetic models knockout β-glucocerebrosidase CGN and dopaminergic cells to characterize the role of β-glucocerebrosidase, please justify??
Answer: Here we aim to dissect the molecular mechanism linking glucosylceramide accumulation with the onset of neuronal damage and to understand if it occurs in a specific neuronal population. In our opinion, the pharmacological inhibition of GCase represents the best way to address this for the following reasons: KO mice for GCase die within 24 hours after birth. Nowadays conditional KO mice for GCase are available, generating only neurons depleted for the enzyme. However, since the enzyme is depleted starting from the progenitors, this could affect also the neuronal development, altering the interpretation of the results, since more pathways can contribute to the observed phenotype. We agree that it could better recapitulate the pathology, but, in our opinion, it does not permit to obtain information on a specific molecular mechanism linking glucosylceramide accumulation and neuronal damage.
Concerning the possibility to use genetically KO dopaminergic neurons, the problem consists in the methodological approach that is necessary to use. In particular, accordingly to Deleidi et al. (Schöndorf DC, Ivanyuk D, Baden P, Sanchez-Martinez A, De Cicco S, Yu C, Giunta I, Schwarz LK, Di Napoli G, Panagiotakopoulou V, Nestel S, Keatinge M, Pruszak J, Bandmann O, Heimrich B, Gasser T, Whitworth AJ, Deleidi M. The NAD+ Precursor Nicotinamide Riboside Rescues Mitochondrial Defects and Neuronal Loss in iPSC and Fly Models of Parkinson's Disease. Cell Rep. 2018 Jun 5;23(10):2976-2988. doi: 10.1016/j.celrep.2018.05.009. PMID: 29874584), in order to silence GBA with high efficiency in order to mimic the pathology, this should be done at the level of iPSCs to obtain stable KO colonies. Nevertheless, this process allows to select cells that are adapted to the lack of the enzyme and are able to survive in this condition and in our opinion this is not the best model to address our question.
Comment: What was the reason behind using 7, 14 days treatment of CBE in CGN cells and 14, 29 days CBE treatment in iPSC-derived dopaminergic neurons? If so then why did the authors not perform 14, 29 days treatment of CBE in CGN cells and 7, 14 days treatment of CBE in iPSC-derived dopaminergic neurons?
Answer: The reason of the different time of treatment of CGNs and iPSCs-derived dopaminergic neurons is exclusively related to the possibility to maintain in culture the two neuronal populations for different periods. Indeed, the iPSCs-derived dopaminergic neurons could be maintained in culture for very long periods, while CGNs are primary neuronal cultures that undergo senescence after 29 days. In addition, the time of treatment has been decided after preliminary experiments aimed to define the early time-point where neuronal degeneration starts and the time point with established neurodegenerative phenotype.
Comment: It would be good if the authors provide a graphical abstract that will draw a great overview of their complete hypothesis?
Answer: As suggested by the reviewer and by the editor we provided a graphical abstract.
Comment: Why the authors did not perform toxicity studies of GlcSph in iPSC dopaminergic neurons (Fig. S7)?
Answer: We added the experiment suggested by the reviewer and we found that DA neurons are sensible to the higher concentration of GlcSph. We modified the text in order to highlight also this aspect. However, the concentration of 10uM of GlcSph is extremely high with respect to that measured in the different tissues and biological fluids of patients affected by GD.
Comment: Fig. S8a, Ganglioside pattern experiment; CBE treatment the GM3, GQ1b, GD1a in CBE treatment the appearing bands are not cleared, The reviewer would like to suggest authors repeat this experiment? Similarly, repeat the experiment for DA neurons.
Answer: As emerge from the HPTLC and as expected, the two cell lines are characterized by different ganglioside pattern, even if both resemble the typical neuronal ganglioside composition due to the presence of complex gangliosides. It is important to consider that the ganglioside pattern was evaluated by the metabolic labelling at the steady state of all the cell sphingolipids using tritiated sphingosine and radioactive lipids were visualised by the digital autoradiograph TRacer Betaimager, BioSpace and quantified using the M3 Vision software biospacelab. This instrument records each beta emission evocated in a single space; this means that the quantification of radioactive spots is not a densitometric analysis but is the sum of all the measured event. For this reason, even if the band is not clearly distinguishable, the quantification is precise. Indeed, this is the result of three independent experiments and the image is only representative and similar to the others obtained.
Comment: Appreciate the results of LAMP1 western blot, immunofluorescence (DA neurons), and Electron microscopy. Please repeat the western blot experiment for LAMP1 in both CGN and DA neurons? It would be good if the author also please check the p62 and LC3-II/I expression levels in both the CGN and DA neurons with CBE treatment?
Answer: we agree with the reviewer’s comment and we substituted the LAMP1 western blot for GCNs with a clearer one. The experiments of the endolysosomal compartment were performed only to justify the effect of the aberrant accumulation of glucosylceramide. Nevertheless, these data and those related to p62 and LC3-II were already reported in literature (Schöndorf, D., Aureli, M., McAllister, F. et al. iPSC-derived neurons from GBA1-associated Parkinson’s disease patients show autophagic defects and impaired calcium homeostasis. Nat Commun 5, 4028 (2014)) and for this reason we decided to do not address it in this paper, which already contain a big amount of data.
Comment: It would be more interesting if the authors check the alpha-synuclein accumulation propensity in β-glucocerebrosidase inhibited CGN and DA neurons?
Answer: we agree with the reviewer, but the study related to the oligomerization of alpha synuclein requires a more complex in vitro model that consists in brain or midbrain organoids. These data have been already obtained and inserted in a different publication that is under submission.
Comment: Please improve the discussion part and check references must be as per journal standard.
Answer: We improved the discussion part of our manuscript. All the changes were reported in track modality.
Comment: Please avoid irrelevant self-citations if not relevant to study and check plagiarism before submission.
Answer: We followed the suggestion of the reviewer.
Comment: In Fig. 1, Harsh condition, the degree of GBA1 deficiency in PD was 30-40% deficiency of GBA1 enzyme activity.
Answer: we agree with the reviewer comment, however here we want to investigate the impact of a general GCase deficiency and the consequent glucosylceramide accumulation on the neuronal homeostasis. For this reason, and to recapitulate the neuronal phenotype that occurs during the entire life of patients in a reasonable timing, we decided to complete blocked GCase activity. However, even if we are convinced that this model does not recapitulate GBA-PD, Mazzulli and Sindransky supposed that also in GBA-PD the complex alpha synuclein-GlcCer induces a competitive inhibition of GCase activity obtaining a low residual activity of the enzyme (Mazzulli, J.R.; Xu, Y.H.; Sun, Y.; Knight, A.L.; McLean, P.J.; Caldwell, G.A.; Sidransky, E.; Grabowski, G.A.; Krainc, D. Gaucher disease glucocerebrosidase and alpha-synuclein form a bidirectional pathogenic loop in synucleinopathies. Cell 2011, 146, 37-52, doi:10.1016/j.cell.2011.06.001).
Comment: Page 2, line 54, substantia nigra pars compactaà not italicized
Answer: We did it.
Comment: Page 2, line 54, PMID: 34644545 should be cited for the finding.
Answer: we cited the publication.
Comment: Page 2, lines 65-67, PMID: 29311330 should be cited for the finding.
Answer: we cited the publication.
Comment: Page 2, lines 69-71, the in vitro models of GCase deficiency using CBE treatment in murine cerebellar granule neurons and iPSCs-derived DA neurons do not represent GBA1-associated PD.
Answer: The use of neurons different from dopaminergic ones allowed us to understand if the molecular mechanism is conserved among the different neuronal population of the brain. This is an important aspect to justify a general impairment of CNS of Gaucher type 2 patients and the diffusion of the neurodegeneration in GBA-PD patients during the evolution of the pathology.
Comment: Page 3, line 139, No information for iPSCs. No information for review and approval from Institutional Stem Cell Research Oversight Committee to use human stem cell
Answer: Approval to use human stem cell has been added to method section.
Comment: Page 3 and 4, more characterization data are required for iPSCs and DA neurons and should be presented. Characterization of iPSC-1) Immunocytochemical and Western blot analysis of the expression of endogenous pluripotent markers; AP, SSEA4, NANOG, TRA-1-60, OCT4, and SSEA3, 2) Karyotypes, 3) If possible Teratoma assay.
Characterization of DA neurons-1) immunocytochemical and Western blot analysis of the expression of the mature neuronal markers; TH, TUJ1, MAP2, PITX3, Nurr1, and FOXA2, 2) Electrophysiological properties such as DA levels from HPLC, Spontaneous firing and action potentials, voltage-gated K+ and Na+ currents
Answer:
We thank the reviewer for the comments. We added the characterization of iPSCs in the supplementary materials as suggested.
Regarding DA neurons, we used a well-established protocol for the differentiation of iPSCs into dopaminergic neurons that shows high efficiency of differentiation. Specifically, the protocol by Zhang et al. (2014) is a slight modification of the protocol described by Kriks et al. 2011, both having a high number of citations. In Supplementary Figures 3 and 4, we show immunoreactivity for TH, TUJ1 and MAP2, markers of proper neuronal and dopaminergic differentiation. In Figure 3 we demonstrate the expression of mature neuronal proteins (MAP2, neurofilament, tau, PDS95), with and without CBE treatment. We did not test the expression of PITX3, NURR1, or FOXA2 since these are markers of floor-plate dopaminergic precursors expressed during differentiation, which were tested in the reference papers of the protocol we used (Kriks et al., 2011; Zhang et al. 2014). We did not perform electrophysiological characterization of DA neurons generated from the line used in this work. However, we applied the same protocol we used in our previous publication (Monzio Compagnogni et al., 2018), in which we performed electrophysiology on DA neurons and showed the presence of evoked action potentials, spontaneous firing, and inward and outward currents
Comment: The reviewer encourages you to use GBA1 as a nomenclature.
Answer: We did it as suggested
Comment: Changes in mRNA should be informative
Answer: We agree with the reviewer. We didn’t check it because we think that presenting data related to the protein expression could be informative as well.
Comment: In Sup fig. 2, Quantification of TH positive neurons is required.
Answer: We quantified the TH positive neurons and added the information in the text.
Comment: Page 6, line 271, GCase activity should be also measured in lysosomal fraction.
Answer: The methods to isolate lysosomes are based on subfractionation experiments that are influenced by the density of the organelles. Since CBE treated cells are characterized by lysosomes engulfed in lipids such as glucosylceramide, the possibility to isolate them with high grade of purification is strongly compromised. In addition, our intent was to demonstrate the efficacy of CBE in blocking GCase activity, for this reason we performed the assay on the total cell lysate.
Comment: In Fig. 1, 4% GBA1 enzyme activity is detected, Why is that? Is that GBA2 enzyme activity?
Answer: we can exclude that the measured activity is due to GBA2 since we block its activity using the AMP-DNM inhibitor. Indeed, as control, each enzymatic assay was performed also in presence of both CBE and AMPDNM and no residual activity was found.
Comment: In Fig. 2, the reviewer is not clear why the author conducted the experiments (move to supplementary data).
Answer: in our opinion this a central figure of the paper since it demonstrates the onset of the neurodegenerative phenotype and that our models could be used to investigate the link between the lysosomal accumulation of GlcCer and the neurodegeneration.
Comment: Page 8, line 412, the subtitle is overstated and Page 10, lines 466-468, the conclusion is also overstated. The presented data do not represent the neurodegenerative phenotype observed in GCase-related pathologies.
Answer: Relating to the comment about the subtitle, the data presented in this paragraph support that CBE treatment induces the degeneration of the neurons, and this is the reason why we think that the title is appropriated. Regarding the Page 10, lines 466-468, we agree with the reviewer and we removed the sentence comparing the neurodegenerative phenotype observed with that found in GCase-related pathologies.
Comment: Page 10, lines 483-484, the sentence should be rewritten
Answer: We rewrote the sentence as following: “The role of GlcSph in the onset of cell toxicity is widely discussed in literature [24-26], therefore we decided to investigate its effect on the viability of CGN and DA neurons by its exogenous administration”.
Comment: Page 10, lines 487-488, the authors should discuss why the levels of Ceremide were not changed in the DA neurons in discussion.
Answer: According to the reviewer comment with implemented the discussion with the following sentence: “In addition, we can exclude the toxic effect of another sphingolipid known to be involved in the activation of the apoptotic process, the ceramide. In particular, we found that its level decreased in CGN, probably because cells do not activate a compensatory mechanism involved in its de novo biosynthesis in order to deal with the reduced availability of sphingosine due to the block of glucosylceramide catabolism. Conversely, we can speculate that such mechanism was activated in DA neurons, since the levels of ceramide are the same in treated and untreated neurons”.
Comment: In Sup Fig. 8. the authors should discuss why there is a difference between granule neurons and DA neurons.
Answer: The differences in term of the ganglioside pattern between CGN and DA neurons are expected since they are two different subpopulation of neurons. Up to now we do not have any explanation on why the CBE treatment produces a different effect on the ganglioside pattern in the two different cell lines. However, as reported in the discussion section, the accumulation of gangliosides is a typical feature of lysosomal storage disorders and it is due to a general lysosomal impairment.
Comment: In Sup fig. 8, standard maker for HPTLC is missing
Answer: Accordingly, we provided a separated file containing the original HPTLC with the standards.
Comment: In Fig 4, validation study is required in a few proteins identified from Masspec
Answer: We inserted the proteomic data as a database available for the scientific community in order to address new studies in the field. For this reason, we do not proceed with the validation of the proteomic data.
Comment: In Fig 5, The reviewer does not see change in Lamp1 in both cells
Answer: we agree with the reviewer concerning the western blot of CGN and we substituted it with a new one. However, the data reported in the graph are the results of a quantification performed on three different experiments. In addition, to confirm the changes in LAMP1 expression, we included also immunofluorescence analysis.
Comment: Please use a dot plot in all data
Answer: We changed the graphs as suggested.
Comment: In Sup Fig. 9C, should be quantified for lysosome biogenesis (number and size). TFEB levels would be informative
Answer: As suggested by the reviewer we quantified lysosomes number and size and we also added information about TFEB levels.
Comment: Page 14, lines 603-604, HPTLC image for extracellular GlcCer should be presented
Answer: Accordingly, HPTLC image for extracellular GlcCer has been added to figure 6.
Comment: In Fig. 7 Standard marker is required for the TLC.
Answer: Accordingly, we provided a separated file containing the original HPTLC with the standards.
Comment: Page 16, line 655, check the references.
Answer: we did it.
Comment: Page 17, lines 724-725, it looks wrong. GlcCer is accumulated in GBA1-PD iPSC derived DA neurons (PMID: 29311330).
Answer: We agree with the reviewer comment and we modified the sentence. However, the iPSCs-derived DA neurons even if generated from GD patients, do not show the same accumulation of GlcCer found in GD patient’s brains.
Comment: Fig. 8 Standard marker is required for the TLC.
Answer: Accordingly, we provided a separated file containing the original HPTLC with the standards.
Comment: What is the syn levels, Is alpha-synuclein oligomer observed? Does CBE increase the half-life of alpha-synuclein ?
Answer: We didn’t check it. The study related to the oligomerization of alpha synuclein requires a more complex in vitro model which consists in the brain or midbrain organoids. These data are already obtained and inserted in a different publication which is under submission.
Reviewer 2 Report
In this study (β-glucocerebrosidase deficiency activates an aberrant lyso- 2 some-plasma membrane axis responsible for the onset of neu- 3 rodegeneration), Lunghi et al. performed the CBE-induced GCase targeting on CGN derived neurons from pups of WT (C57BL/6) mice and the human DA neurons differentiated from iPSCs derived from a healthy subjects to confirm whether the GCase deficiency and the resultant excess cell making of the GlcCer contribute to the neurodegeneration in Gaucher disease. The findings of this study suggest that β-glucocerebrosidase defects impairs the lysosomal compartment, establishing a lysosome-plasma membrane axis responsible for modifications in the plasma membrane architecture and possible alterations of intracellular signaling pathways, leading to neuronal damage.
The Manuscript is well written and the findings of this study are extremely useful for the field. I will strongly recommend this MS for publication without any further modification
Author Response
Reviewer #2: We thank the reviewer for the kind comment and for appreciate our work
Round 2
Reviewer 1 Report
I think that the authors have adequately addressed the comments made by the reviewer in the revised version of the manuscript. Therefore, I have no further comments.